# IOHFuseLM: Multimodal Forecasting of Sparse Intraoperative Hypotension Events Powered by Language Model

## Abstract

Intraoperative hypotension (IOH) is a common complication of general anesthesia and is strongly associated with adverse outcomes such as myocardial injury and increased mortality. Despite its significance, IOH prediction is hindered by event sparsity and the challenge of integrating heterogeneous static attributes and dynamic physiological signals. In this paper, we propose a multimodal language model framework **IOHFuseLM**. To accurately identify and differentiate sparse hypotensive events, we leverage a two-stage training strategy. The first stage involves domain adaptive pretraining on IOH physiological time series augmented through diffusion methods, thereby enhancing the sensitivity to patterns associated with hypotension. Subsequently, task fine-tuning is performed on the original clinical dataset to further enhance the ability to distinguish normotensive from hypotensive states. To enable multimodal fusion for each patient, we align structured clinical descriptions with the corresponding physiological time series at the token level. Such alignment enables the model to capture individualized temporal patterns alongside their corresponding clinical semantics. In addition, we transform static patient attributes into structured text to enrich personalized information. Experiments on two intraoperative datasets and one arrhythmia dataset demonstrate that IOHFuseLM outperforms baselines in IOH identification and generalizes effectively to abnormal heartbeat detection, underscoring its potential as a versatile solution across physiological domains. Our code is publicly available to promote reproducibility at `https://anonymous.4open.science/r/IOHFuseLM-C5A4`.

## 1 Introduction

Intraoperative hypotension (IOH) is a common complication during surgery and has been associated with adverse postoperative outcomes (Kouz et al., 2020; Johnson et al., 2016), including myocardial injury (Van Waes et al., 2016) and increased mortality (Wijnberge et al., 2021). Given its high prevalence and substantial implications (Wesselink et al., 2018), the development of accurate IOH prediction models has become a critical objective in perioperative monitoring (Saasouh et al., 2024).

Conventional approaches to IOH prediction primarily rely on physiological features such as arterial blood pressure (Hatib et al., 2018), and increasingly incorporate deep learning models to capture temporal dependencies in time series data (Lee et al., 2021). These methods typically leverage convolutional neural networks to extract local series patterns (Jeong et al., 2024), or employ recurrent and attention-based architectures to model sequential dynamics (Kwon et al., 2018), achieving moderate performance gains. However, most existing methods either focus exclusively on physiological time series (Jeong et al., 2019; Cheng et al., 2024) or adopt simple feature-level fusion strategies by concatenating static patient attributes (Lu et al., 2023), without fully modeling the semantic and contextual complexity of individual patients.

Recent advances in deep learning for time series forecasting have led to notable progress across diverse domains, with models such as LSTM (Graves & Graves, 2012), Transformer-based (Zhang & Yan, 2023; Liu et al., 2023; Wang et al., 2024), and MLP-based architectures (Zeng et al., 2023; Yi et al., 2023) demonstrating strong capabilities in modeling temporal dynamics. Beyond deterministic models, diffusion-based generative approaches are effective for time series analysis (Yuan & Qiao,

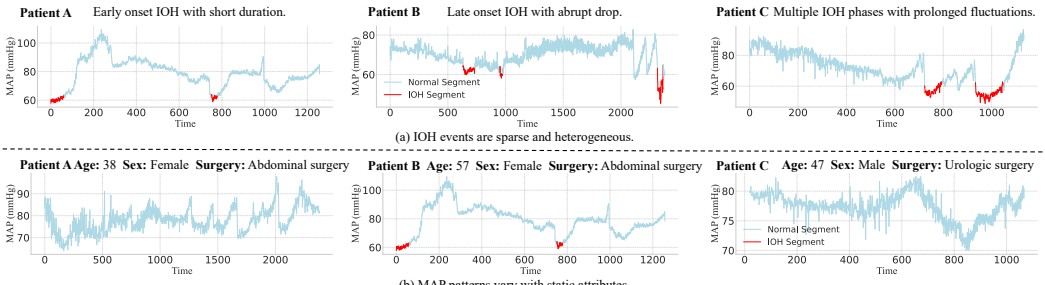

Figure 1: (a) IOH events are sparse and exhibit substantial inter patient variability in onset time, duration, and waveform morphology. (b) MAP series vary significantly across static attributes including age groups, genders and surgery types.

2024; Liu et al., 2024a) and data augmentation via realistic sample synthesis (Trabucco et al., 2023). More recently, language models (Zhou et al., 2023; Jin et al., 2023) have expanded the field of time series forecasting by enabling cross-modal representation learning and effectively aligning textual and temporal features.

Despite substantial progress, predicting IOH remains challenging (Yang et al., 2024). As shown in Figure 1 (a), IOH events are sparse, brief, and highly variable in onset time, waveform morphology, and temporal dynamics. Figure 1 (b) further shows that MAP fluctuations vary significantly across patients due to factors such as age and type of surgery, making it difficult for models to generalize across diverse populations. Effective IOH modeling thus requires capturing critical temporal patterns while jointly integrating static patient attributes (Temesgen et al., 2021).

To address the challenge posed by sparse IOH events, we propose a multimodal language model framework IOHFuseLM, which integrates static patient attributes with dynamic physiological series. The training process consists of two stages. First, domain adaptive pretraining is conducted on a dataset augmented with a diffusion strategy to capture a diverse range of fine-grained patterns. This is followed by task fine-tuning using a customized loss function that improves the sensitivity to IOH-related abnormalities. Static attributes are transformed into clinically informed descriptions, enabling cross-modal alignment through token level interaction for precise semantic fusion.

Our main contributions are summarized as follows:

- We propose IOHFuseLM, a novel multimodal language model framework for IOH prediction. The model is trained using a two-stage paradigm: domain adaptive pretraining on a diffusion-augmented physiological dataset, followed by task fine-tuning on real intraoperative records.

- We develop a clinically informed multimodal fusion strategy that aligns static patient context with temporal physiological series by converting patient attributes into structured clinical text and aligning it at the token level with physiological series.

- Experiments on three real-world clinical datasets, including a curated dataset of raw intraoperative blood pressure recordings, demonstrate that our method consistently outperforms competitive baselines and shows potential for generalizability across clinical scenarios.

## 2 RELATED WORK

**Intraoperative Hypotension Forecasting.** Modeling intraoperative arterial pressure has emerged as a key strategy for early prediction of intraoperative hypotension (IOH), enabling timely clinical interventions and improved patient safety. Early efforts primarily focused on high fidelity arterial pressure series, leading to the development of the Hypotension Prediction Index (Hatib et al., 2018). Subsequent machine learning approaches, including ensemble methods (Cherifa et al., 2020) and gradient boosting techniques (Kendale et al., 2018), integrated both preoperative and intraoperative variables. However, these models often treated each data point in isolation, overlooking the intrinsic temporal dependencies. To address this limitation, deep learning architectures, including recurrent neural networks (Jeong et al., 2019) and attention-based models (Lu et al., 2023), were introduced to better capture sequential patterns. More recently, interpretable models (Ritter et al., 2023; Hwang et al., 2023) have improved clinical utility, although they still depend on predefined features and

structured inputs. Meanwhile, the frequency-domain perspective (Moon et al., 2024) has also been explored. While existing IOH prediction methods have made considerable progress, most are grounded in either biomarker identification or deep learning models that lack the capacity to align patient specific clinical narratives with the evolving temporal dynamics of physiological series. Despite progress, existing IOH prediction methods still lack the capacity to bridge multimodal disparities and effectively model personalized, temporally evolving risk patterns.

**Time Series Forecasting.** Time series forecasting plays a pivotal role in many domains. Classical statistical models, including ARIMA (Ariyo et al., 2014), often struggle to capture the complex dynamics of physiological series with high dimensionality and nonlinearity. Deep learning models, such as long short term memory networks (Graves & Graves, 2012) and gated recurrent units (Dey & Salem, 2017), have demonstrated strong capabilities in modeling temporal dependencies over extended time horizons by leveraging gated mechanisms. In recent years, Transformer-based architectures (Zhou et al., 2021; Wu et al., 2021) have achieved notable progress in time series forecasting. For instance, PatchTST (Nie et al., 2022) introduces patch-level embeddings, providing a principled approach to tokenizing time series. MLP-based models (Zeng et al., 2023; Ekambaram et al., 2023) and CNN-based models (Luo & Wang, 2024; Wang et al., 2023) have also shown competitive performance, effectively capturing temporal dependencies.

In addition to architectural advancements, generative modeling has emerged as a promising paradigm for time series forecasting, with diffusion-based approaches gaining increasing attention. Recent models (Tashiro et al., 2021; Shen & Kwok, 2023) effectively capture complex temporal dynamics by iteratively denoising noise-perturbed series through learned reverse processes. At the same time, large language models have exhibited increasing potential in time series modeling (Zhou et al., 2023; Jin et al., 2023; Pan et al., 2024). Through pretraining and instruction tuning, LLMs are capable of generalizing forecasting capabilities across a wide range of tasks and domains, thereby enabling more flexible and adaptive series understanding. These advancements establish a solid foundation for developing unified and generalizable time series forecasting frameworks that combine high representational capacity with strong adaptability to the intricate dynamics characteristic of IOH prediction scenarios.

## 3 PRELIMINARIES

**Definition of Intraoperative Hypotension.** Intraoperative hypotension (IOH) is defined according to clinically established thresholds. An IOH event is identified when the mean arterial pressure (MAP) remains below 65 mmHg for at least one continuous minute (Sessler & Khanna, 2018; Sessler et al., 2019). Systolic blood pressure (SBP) and diastolic blood pressure (DBP) denote the maximum and minimum pressures within a cardiac cycle, respectively. The MAP (Meaney et al., 2000), a critical indicator of cardiac output and systemic vascular resistance (Magder, 2018), is calculated as:

$$\text{MAP} = \frac{\text{SBP} + 2 \times \text{DBP}}{3}, \tag{1}$$

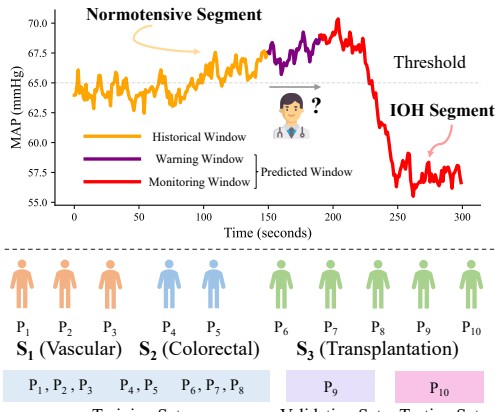

Figure 2: Top: Temporal segmentation for IOH prediction. The MAP curve is divided into historical window (orange), warning window (purple), and monitoring window (red). Bottom: Patients are split by procedure into training, validation, and test sets to ensure subject independence and prevent data leakage.

**Series Instance Construction.** Given a historical window of length $L$, the model predicts a future MAP series of length $T$, as illustrated in Figure 2. To prevent label leakage and enable realistic forecasting, instances with historical windows overlapping IOH episodes are excluded. To mitigate class imbalance and capture temporal dynamics, we adopt an adaptive slicing strategy: negative instances are sampled at regular intervals $\Delta_{\text{Normal}}$ to reduce redundancy, while positive instances linked to IOH are sampled more frequently at intervals $\Delta_{\text{IOH}}$ to ensure adequate coverage of critical transitions.

**Surgery Aware Subject Splitting.** To ensure realistic generalization, we employ a subject independent split by assigning each patient exclusively to the training, validation, or test set, as shown in Figure 2. Patients are grouped according to their surgery type, and each group is assigned to only one data partition. This stratification helps maintain a balanced distribution of surgery types across splits, thus mitigating any distributional shifts caused by surgery specific hypotension risks. This strategy also prevents the memorization of subject-specific patterns and reflects real world deployment scenarios. Moreover, it enables a clear separation of static attributes and temporal waveform data across splits, which facilitates model generalization to unseen individuals.

**IOH Event Evaluation.** The ground-truth label for each timestamp is assigned based on whether the subsequent one-minute MAP series remains continuously below 65 mmHg. A predicted IOH event is assigned if more than 60% of the forecasted MAP values within the same one-minute window fall below this threshold. Model performance is evaluated using pointwise and instance-level metrics to capture the effectiveness in detecting IOH events.

# 4 METHODOLOGY

In this section, we describe the framework IOHFuseLM for intraoperative hypotension (IOH) prediction. As shown in Figure 3, the framework consists of four components: personalized clinical description generation, multi-scale trend-residual diffusion augmentation, domain adaptive pretraining, and task fine-tuning. IOHFuseLM is built on GPT-2 architecture (Radford et al., 2019).

## 4.1 PERSONALIZED CLINICAL DESCRIPTION GENERATION

To incorporate static features effectively, we propose a template-guided Personalized Clinical Description Generation (PCDG) module that fuses physician recommendations, institutional expertise, and the literature **to produce structured patient-specific descriptions**. By leveraging curated clinical knowledge and retrospective studies (Maleczek et al., 2024; Saasouh et al., 2024), the module yields individualized narratives that model cross attribute semantics with domain alignment, minimizing feature engineering and projection-induced modality gaps. Details are provided in Appendix D.

For each patient $p_j$, the personalized clinical description is defined as:

$$\boldsymbol{d}_j = \phi(a_j, g_j, s_j). \tag{2}$$

Here, $\phi$ represents GPT-4o, which generates $\boldsymbol{d}_j$ from the static attributes $(a_j, g_j, s_j)$ under a predefined medical template. To improve clinical relevance, the tokenizer is augmented with hormone- and surgery-related terms. For patient $j$, define $\mathcal{T}_j$ as the set of MAP physiological series instances. Accordingly, the paired dataset is defined as:

$$\mathcal{X}_1 = \big\{ (\boldsymbol{d}_j, \boldsymbol{x}_i) \, \big| \, j \in [N], \, \boldsymbol{x}_i \in \mathcal{T}_j \big\}. \tag{3}$$

## 4.2 MULTI-SCALE TREND-RESIDUAL DIFFUSION AUGMENTATION

To alleviate the challenge of scarce IOH cases, which hampers accurate modeling of hemodynamic series, we propose the Multi-Scale Trend Residual Diffusion Augmentation (MTRDA) module. This approach enhances the representation and generation of **sparse MAP series**, particularly those containing IOH events. MTRDA improves the ability to learn both broad temporal patterns and fine-grained variations within hypotensive intervals.

Adaptive smoothing is initially performed on the MAP series defined over the historical window to extract underlying trends. Specifically, we employ a set of centered sliding average filters with predefined odd-length window kernels $\mathcal{S} = \{w_1, w_2, \ldots, w_{|\mathcal{S}|}\}$. For each scale $w_s \in \mathcal{S}$, the smoothed series $\boldsymbol{x}_i^{(s)}$ is computed as:

$$\boldsymbol{x}_{i,t}^{\langle s \rangle} = \frac{1}{w_s} \sum_{\tau=-\lfloor w_s/2 \rfloor}^{\lfloor w_s/2 \rfloor} \boldsymbol{x}_{i.t+\tau}, \tag{4}$$

where $\lfloor \cdot \rfloor$ denotes the floor operator. Boundary values are handled via symmetric padding. The final multiscale trend estimate is obtained by averaging across all scales:

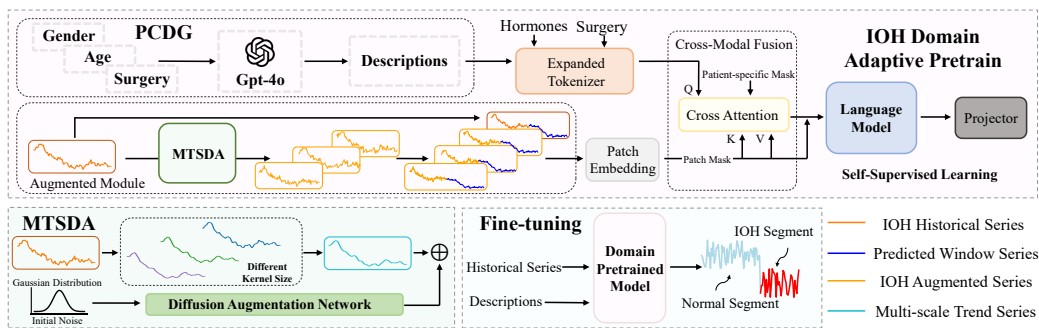

Figure 3: Illustration of our framework. MTRDA decomposes MAP series into multi-scale, and enhances IOH historical series via diffusion-based augmentation. IOH domain adaptive pretraining aligns the augmented IOH series and clinical descriptions through dual-masked cross-attention under a self-supervised objective. Task fine-tuning incorporates labeled normotensive and IOH series with an IOH-specific MSE loss to refine event detection.

$$\boldsymbol{x}_{i,\text{trend}} = \frac{1}{|\mathcal{S}|} \sum_{s=1}^{|\mathcal{S}|} \boldsymbol{x}_i^{\langle s \rangle}, \tag{5}$$

$$\boldsymbol{x}_{i,\text{residual}} = \boldsymbol{x}_{i,1:l} - \boldsymbol{x}_{i,\text{trend}}, \tag{6}$$

where $\boldsymbol{x}_{i,\text{trend}}$ and $\boldsymbol{x}_{i,\text{residual}}$ represent the trend and residual components of the series, respectively. Short windows capture rapid fluctuations indicative of oscillatory patterns, while long windows reveal sustained trends linked to patient status. This multiscale smoothing strategy **preserves structural patterns** $X_i^{\textbf{trend}}$ across temporal levels and facilitates early IOH detection. The residual component $\boldsymbol{r}_i$ retains detailed variations reflecting subtle physiological dynamics.

To enhance the residual component $\boldsymbol{x}_{i,\text{residual}}$ by preserving the overall MAP trend while **enriching fine-grained fluctuations**, MTRDA incorporates a diffusion-based generative mechanism that learns to reconstruct and refine the residual series through iterative denoising.

$$\boldsymbol{x}_{i,\text{residual}}^{(k)} = \sqrt{\bar{\alpha}_k}\,\boldsymbol{x}_{i,\text{residual}}^{(0)} + \sqrt{1 - \bar{\alpha}_k}\,\boldsymbol{\epsilon}, \tag{7}$$

where $k$ denotes the diffusion step, $\bar{\alpha}_k$ denotes the cumulative product of the noise schedule coefficients, and $\boldsymbol{\epsilon} \sim \mathcal{N}(0, \boldsymbol{I})$ is standard Gaussian noise.

$$L_{\text{ELBO}} = \mathbb{E}_{\boldsymbol{x}_{i,\text{residual}}, k} \left\| \boldsymbol{x}_{i,\text{residual}}^{(0)} - f_\theta\left(\boldsymbol{x}_{i,\text{residual}}^{(k)}, k\right) \right\|^2. \tag{8}$$

The diffusion augmentation network $f_\theta$ comprises three modules. During training, the embedding module encodes the residual series into a high-dimensional latent space using multilayer perceptrons and learnable positional encodings, with diffusion step $k$ integrated via sinusoidal encodings (Ho et al., 2020; Gu et al., 2022) and Adaptive Layer Normalization (AdaLN) (Guo et al., 2022). The encoded features are passed to a lightweight denoising decoder composed of stacked linear layers and normalization blocks, which iteratively refine the residual series while reducing computational overhead. A projection layer then maps the refined representation back to the residual space and combines it with the trend component $\boldsymbol{x}_{i,\text{trend}}$ to generate the output and compute the training loss. Further details of the augmentation network are provided in Appendix C, and the formal derivation is presented in Appendix I. For each original series, initial noise is sampled from a Gaussian distribution and passed through the trained network $f_\theta$ together with the trend component $\boldsymbol{x}_{i,\text{trend}}$, generating $H$ augmented MAP series. These are denoted as $\boldsymbol{X}' = \{\boldsymbol{x}'^{(1)}, \dots, \boldsymbol{x}'^{(H)}\}$ and preserve both transient anomalies and subtle fluctuations. We then construct the extended dataset as:

$$\mathcal{X}_2 = \mathcal{X}_1 \cup \left\{ \left(\boldsymbol{d}_j, \boldsymbol{x}_{i,1:l}'^{(h)} \oplus \boldsymbol{x}_{i,l+1:l+t}\right) \mid j \in [N],\ \boldsymbol{x}_i \in \mathcal{T}_j, h \in [H] \right\}. \tag{9}$$

This reconstruction process captures fine-grained residual patterns within the historical IOH window, thereby enhancing the fidelity and informativeness of sparse IOH series.

### 4.3 DOMAIN ADAPTIVE PRETRAINING

Pretraining has demonstrated effectiveness in time series analysis (Ma et al., 2024). Our objective is to empower a language model to **comprehend temporal IOH dynamics**, while **enabling cross-modal fusion** between physiological time series and patient-specific clinical descriptions. To harness this potential in IOH forecasting, we propose a domain adaptive pretraining strategy that aligns personalized clinical context with IOH patterns.

Specifically, each input pair $(\hat{\boldsymbol{x}}_i, d_j)$ is sampled from $\mathcal{X}_2$. To enable modality alignment, the MAP series $\hat{\boldsymbol{x}}_i$ is segmented into fixed-length patches of size $p$, which are linearly projected into MAP patch tokens. A random masking ratio $R$ is applied to the resulting tokens to enhance representation learning. The corresponding clinical description $d_j$ is tokenized using expanded language model tokenizer, yielding the text token $T_i$ with a maximum length of $\eta$.

To enable selective fusion of patient-specific textual and physiological features, we construct a patient-specific attention mask $\boldsymbol{M}_i$. Specifically, we define two binary vectors: the first is a vector $\mathbf{1}_{\frac{l+t}{p}}$, whose length matches the number of series tokens, with all elements set to active. The second is a binary vector $\boldsymbol{m}_i$ of length $\eta$, corresponding to the text token $T_i$, where active elements represent valid tokens and inactive elements represent padding positions. Additionally, we define an all-active vector $\mathbf{1}_\eta$ of length $\eta$. These masks are combined using elementwise logical conjunction to form the joint attention mask $\boldsymbol{M}_i$, defined as:

$$\boldsymbol{M}_i = \mathbf{1}_{\frac{l+t}{p}} \cdot (\mathbf{1}_\eta - \boldsymbol{m}_i)^\top, \tag{10}$$

$$\text{Attention}(\boldsymbol{Q}, \boldsymbol{K}, \boldsymbol{V}) = \text{softmax}\left(\frac{\boldsymbol{Q}\boldsymbol{K}^\top}{\sqrt{d}} - \lambda \cdot \boldsymbol{M}_i\right) \boldsymbol{V}, \tag{11}$$

where $\lambda$ is a large constant to suppress attention to semantically misaligned regions. We adopt the token level alignment mechanism (Liu et al., 2024b), which aligns the text token $T_i$ with the patch token of series $\hat{\boldsymbol{x}}_i$. The query matrix $Q$ is derived from the tokenized and projected text tokens $T_i$, while the key and value matrices $K$ and $V$ are obtained from the corresponding MAP series tokens. The resulting representations are concatenated with the series tokens and processed by a pretrained language model, followed by a projection linear layer to predict the IOH related MAP series. The model is optimized to minimize the mean squared error (MSE) on the masked positions of the time-series tokens.

This pretraining strategy enables the model to learn semantically meaningful interactions between the IOH related MAP series and the corresponding static patient features. It also facilitates modality alignment between IOH series and patient specific information, thereby providing a stronger language model foundation for subsequent hypotension prediction.

### 4.4 TASK FINE-TUNING

To adapt the pretrained model to the downstream IOH prediction task, the task fine-tuning stage further refines the representations learned after domain adaptive pretraining, thereby enhancing the ability to **distinguish IOH patterns from normotensive fluctuations**.

Each input pair $(d_j, \boldsymbol{x}_i) \in \mathcal{X}_1$ is first processed to derive token level representations that capture instance-specific semantic and physiological characteristics. These representations are integrated with the corresponding series embeddings and then passed to the pretrained model. During task fine-tuning, the series embedding and output projection layers are reinitialized for task adaptation, while all other parameters are initialized from the domain adaptive pretraining stage and jointly optimized to improve temporal sensitivity to IOH dynamics.

To enhance sensitivity to IOH abnormalities, an additional loss term is introduced for the timestamps of IOH series, defined as the MSE on those timestamps and weighted by a hyperparameter $\rho$. The total IOH loss function is given by:

$$\text{Loss} = \text{MSE}_{\text{normal}} + \rho \cdot \text{MSE}_{\text{IOH}}, \tag{12}$$

where $\text{MSE}_{\text{normal}}$ and $\text{MSE}_{\text{IOH}}$ represent the mean squared errors computed over normotensive and hypotensive series, respectively. This task fine-tuning strategy encourages the model to attend to subtle temporal variations indicative of IOH, thereby enhancing predictive performance and facilitating timely clinical intervention.

## 5 EXPERIMENTS

### 5.1 EXPERIMENTAL SETUP

**Datasets.** We use two clinical intraoperative hypotension (IOH) datasets. **Clinical IOH Dataset.** The dataset includes intraoperative records from 6,822 patients, featuring MAP time series resampled at 6 and 10 seconds from arterial blood pressure waveforms, together with patient attributes such as age, gender, and surgery type. After preprocessing, 1,452 recordings were retained. **VitalDB Dataset.** (Lee et al., 2022) This dataset originally consisted of 6,388 ABP recordings. After filtering out low-quality samples, 1,522 recordings were retained for downstream experiments. Both datasets are split by patient into training, validation, and test sets in a 3:1:1 ratio. We use a 15-minute historical window and prediction horizons of 5, 10, and 15 minutes, guided by clinical evidence on IOH predictability (Awad et al., 2022). To further demonstrate the generalization of our framework across diverse scenarios, we validated its generalizability on the **MIT-BIH Arrhythmia Dataset** (Moody & Mark, 2001), a widely used benchmark for abnormal heartbeat detection. Detailed preprocessing steps and dataset statistics are provided in Appendix A.

**Baselines.** We compare our method against six representative time series forecasting models. These include the MLP-based DLinear (Zeng et al., 2023), the Transformer-based PatchTST (Nie et al., 2022), and the frequency domain enhanced Fredformer (Piao et al., 2024). We also include HMF (Cheng et al., 2024), a model specifically designed for intraoperative hypotension prediction, along with two language model-based approaches: GPT4TS (Zhou et al., 2023), built on GPT-2, and TimeLLM (Jin et al., 2023), based on LLaMA2-7B (Touvron et al., 2023).

**Implementation.** To assess IOH prediction performance, we report Mean Squared Error (MSE) and Mean Absolute Error (MAE) on hypotensive timestamps. Discriminative ability is measured by the Area Under the ROC Curve (AUC), and Recall reflects early warning effectiveness. All models are trained with the Adam optimizer (Kingma & Ba, 2014), using a batch size of 8 and an initial learning rate of 0.0001 with a decay factor of 0.75. Experiments are conducted on a NVIDIA RTX 4090 GPU and a server equipped with eight Tesla A100 GPUs. To ensure result reliability, we report averages over three independent runs. Full training configurations are provided in Appendix B.

### 5.2 RESULTS AND DISCUSSION

Table 1: Performance comparison of different models on the Clinical IOH and VitalDB datasets under varying sampling rates. The best result for each metric is indicated in bold.

| Dataset | Historical Window | Sampling (s) | Model | $MSE_{IOH}$ | $MAE_{IOH}$ | Recall | AUC |
|---|---|---|---|---|---|---|---|
| Clinical IOH | 150 | 6 | DLinear | 178.6592 | 10.9190 | 36.61% | 0.6406 |
| | | | PatchTST | 122.2292 | 8.5687 | 63.09% | 0.6948 |
| | | | Fredformer | 99.0389 | 7.7170 | 59.98% | 0.6985 |
| | | | HMF | 114.6592 | 8.3079 | 50.76% | 0.6737 |
| | | | GPT4TS | 119.0686 | 8.4467 | 59.37% | 0.6991 |
| | | | TimeLLM | 133.2503 | 9.2422 | 46.58% | 0.6687 |
| | | | **IOHFuseLM** | **88.9192** | **7.4921** | **74.00%** | **0.7130** |
| | 90 | 10 | DLinear | 127.0857 | 8.6489 | 51.49% | 0.6933 |
| | | | PatchTST | 125.5609 | 8.8750 | 56.84% | 0.7044 |
| | | | Fredformer | 103.5407 | 8.0945 | 53.17% | 0.6935 |
| | | | HMF | 121.1721 | 8.7806 | 51.40% | 0.6853 |
| | | | GPT4TS | 91.4255 | 7.4369 | 62.68% | 0.7309 |
| | | | TimeLLM | 118.6838 | 8.7864 | 50.45% | 0.6913 |
| | | | **IOHFuseLM** | **87.6147** | **7.3933** | **74.46%** | **0.7425** |
| VitalDB | 300 | 3 | DLinear | 92.2800 | 7.3100 | 33.48% | 0.6300 |
| | | | PatchTST | 99.3965 | 7.6942 | 52.62% | 0.6443 |
| | | | Fredformer | 69.5776 | 6.0244 | 49.94% | 0.6640 |
| | | | HMF | 76.5757 | 6.5151 | 49.73% | 0.6501 |
| | | | GPT4TS | 61.7742 | 5.5824 | 57.39% | 0.6885 |
| | | | TimeLLM | 82.3817 | 7.0517 | 30.36% | 0.6068 |
| | | | **IOHFuseLM** | **58.3511** | **5.1251** | **70.10%** | **0.7086** |

**Main Results.** We conduct comprehensive experiments on the Clinical IOH dataset and VitalDB dataset. Results are summarized in Table 1. Experimental results highlight key differences among baseline models. The moderate performance of DLinear reflects limitations in capturing complex temporal dynamics using simple linear decomposition. PatchTST excels in recall and AUC by segmenting series into semantically meaningful patches. Fredformer improves performance by reducing frequency bias but struggles with the high variability of IOH events. HMF extracts temporal features using sliding windows but lacks semantic modeling of baseline blood pressure associated with surgery type, leading to poor generalization across procedures. GPT4TS performs well on high-frequency data, capturing short physiological trends effectively. The fixed parameters of TimeLLM limit its adaptability to distribution shifts in MAP series. IOHFuseLM outperforms others in sparse and high-variability settings by aligning static patient attributes with MAP series and augmenting sparse data with realistic signals, effectively addressing IOH sparsity and variability.

Performance varies distinctly across datasets and sampling rates. VitalDB, with higher IOH event density, generally yields better metrics, particularly for IOHFuseLM, which excels at fine granularity, demonstrating strong temporal pattern extraction capabilities. Clinical IOH, characterized by sparser events, presents greater modeling challenges, yet IOHFuseLM consistently maintains strong performance across coarser sampling intervals. These results emphasize adaptability and the effectiveness of its multimodal context integration and data augmentation strategies.

**Ablation Results.** The following ablated variants are evaluated:

- **IOHFuseLM[1]**: Excludes the clinical description $d_j$, only modeling the MAP time series.

- **IOHFuseLM[2]**: Utilizes the original GPT-2 tokenizer without vocabulary expansion.

- **IOHFuseLM[3]**: Conducts domain adaptive pretraining exclusively on the original dataset $\mathcal{X}_1$, omitting any diffusion-based augmentation.

- **IOHFuseLM[4]**: Removes the domain adaptive pretraining stage and directly applies task fine-tuning on the downstream IOH prediction task.

We conduct a detailed ablation study on the Clinical IOH dataset sampled at 10-second intervals, using a historical window of 15 minutes and predicted horizons of 5, 10, and 15 minutes. Table 2 summarizes the impact of removing each key component from IOHFuseLM. The results confirm that every component is essential for addressing the challenges discussed above. Removing static attributes degrades performance by eliminating personalized priors that help identify MAP trends and variability, reducing the ability to detect abnormal patterns and generalize across populations and surgery types. Excluding the expanded tokenizer weakens the ability to associate clinical terminology with physiological patterns, diminishing cross-modal representation learning. Using only the original dataset $\mathcal{X}_1$ for domain adaptation pretraining, instead of the MTRDA dataset, reduces sensitivity to rare and short-duration IOH episodes, demonstrating the benefit of synthetic variability in mitigating data sparsity. Finally, omitting the domain adaptation pretraining stage leads to consistent performance degradation across all metrics, confirming that prior exposure to IOH patterns enhances generalization under limited supervision.

Table 2: Ablation study of model components on the Clinical IOH dataset.

| Dataset | Model Variant | $\text{MSE}_{\text{IOH}}$ | $\text{MAE}_{\text{IOH}}$ | Recall | AUC |
|---|---|---|---|---|---|
| | IOHFuseLM | **87.6147** | **7.3933** | **74.46%** | **0.7425** |
| | IOHFuseLM[1] | 87.6824 | 7.4604 | 68.62% | 0.7215 |
| Clinical IOH | IOHFuseLM[2] | 95.4979 | 7.8967 | 69.42% | 0.7287 |
| | IOHFuseLM[3] | 107.7359 | 8.3523 | 67.78% | 0.7213 |
| | IOHFuseLM[4] | 98.1118 | 7.9061 | 67.85% | 0.7192 |

**Transfer Results.** To evaluate the generalization ability, we conducted transfer learning experiments on a newly curated cohort of patients with surgical durations between 600 and 1000 seconds, sampled every 6 seconds. The historical and predicted windows were set to 3 and 5 minutes, respectively. The model was pretrained on the Clinical IOH dataset and tested under the settings, focusing on adult patients aged 18–65 years. As shown in Table 3, transfer learning substantially improved performance across all metrics, particularly in recall and AUC. These gains indicate enhanced sensitivity to IOH events and better discrimination under demographic and procedural variability. The results highlight

the effectiveness of domain adaptive pretraining and personalized context integration in enabling effective generalization across diverse clinical scenarios. Further analyses of practicality and field deployment are provided in Appendices G and H.

Table 3: Performance comparison with and without transfer learning on the Clinical IOH dataset.

| Transfer Setting | Historical Window | Predicted Window | MSE$_{IOH}$ | MAE$_{IOH}$ | Recall | AUC |
|---|---|---|---|---|---|---|
| Without Transfer Learning | 30 | 50 | 216.4261 | 14.2554 | 0.00% | 0.5000 |
| Transfer Learning | 30 | 50 | 97.9322 | 9.0675 | 17.65% | 0.5805 |

**Generalization Evaluation to Rare Clinical Events.** To examine the adaptability of our framework beyond intraoperative hypotension, we applied IOHFuseLM to abnormal heartbeat prediction using the MIT-BIH Arrhythmia dataset. The decision threshold was determined by Youden's J statistic to balance recall and specificity. As shown in Table 4, IOHFuseLM achieves lower MSE and MAE and higher recall and AUC than baselines. These results demonstrate that the proposed structure-aware and anomaly-sensitive design enables robust detection of sparse, clinically significant events across diverse physiological domains. With a consistent architecture requiring only minor task-specific adjustments, the framework shows scalability to broader medical forecasting tasks.

Table 4: Performance comparison on abnormal heartbeat detection.

| | IOHFuseLM | | | | | HMF | | | | | TimeLLM | | | | |
|---|---|---|---|---|---|---|---|---|---|---|---|---|---|---|---|
| Model | MSE | MAE | Recall | AUC | Thres. | MSE | MAE | Recall | AUC | Thres. | MSE | MAE | Recall | AUC | Thres. |
| Values | **1.1397** | **0.7676** | **0.3431** | **0.5961** | 0.0129 | 1.2022 | 0.8422 | 0.3234 | 0.5854 | 0.0121 | 1.1642 | 0.7950 | 0.3298 | 0.5718 | 0.0122 |

**Visual Evidence for domain Adaptive Pretraining.** To qualitatively evaluate the effect of domain adaptive pretraining, we visualize MAP forecasts under two representative IOH patterns: one with a gradual decline and the other with a rapid decline. As shown in Figure 4, we compare models trained with and without pretraining using the same total series length $l + t$, where the $l$ are set to 50 ,100 and 150, respectively. In both patterns, the models with pretraining produce predictions that more closely follow the ground truth, especially in their ability to capture downward trends in blood pressure. This advantage is particularly clear in the rapid-decline scenario, where models without pretraining tend to respond more slowly and deviate further from actual MAP values. When the historical length $l$ is sufficiently long, for example 150, the pretrained model produces stable and accurate forecasts, benefiting from richer temporal context and more reliable trend estimation. This suggests that the proposed pretraining method enhances the model sensitivity to temporal changes and improves its ability to recognize IOH-specific patterns. Appendix F provides additional visualization details.

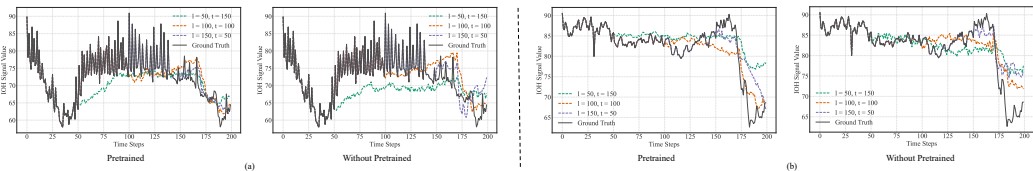

Figure 4: Qualitative comparison of models with and without domain adaptive pretraining under two representative IOH patterns: (a) gradually declining MAP; (b) rapidly declining MAP.

## 6 CONCLUSION

In this work, we introduced IOHFuseLM, a multimodal framework for sparse intraoperative hypotension (IOH) prediction that integrated static patient attributes with dynamic physiological time series data. Evaluations on two real world intraoperative datasets showed that IOHFuseLM consistently outperformed competitive baselines, with particularly strong performance under coarse sampling and sparse event conditions. The proposed approach achieved higher recall and area under the curve (AUC) by effectively capturing patient-specific variability and extracting features with rich temporal and semantic content. Compared to previous methods that relied solely on physiological series or used simple feature concatenation, IOHFuseLM demonstrated better flexibility and generalization through data augmentation during pretraining and structured integration of patient attributes. These results suggest that the model can support personalized and timely clinical monitoring, which may help facilitate early intervention, improve hemodynamic stability during surgery, and reduce the risk of postoperative complications. Moreover, results on MIT-BIH Arrhythmia dataset confirm its applicability to abnormal heartbeat detection and to broader sparse-event critical prediction tasks.

## 7 ETHICS STATEMENT

This study uses two de-identified intraoperative hypotension (IOH) datasets and one publicly available arrhythmia dataset. The Clinical IOH dataset was collected under institutional ethical approval and de-identified according to HIPAA Safe Harbor standards. The VitalDB dataset is released under its official CC BY-NC-SA 4.0 license, and the MIT-BIH Arrhythmia dataset is a widely used community benchmark. No new human or animal experiments were conducted.

## 8 REPRODUCIBILITY STATEMENT

We make significant efforts to ensure reproducibility. Details of dataset preprocessing, training procedures, hyperparameters, and evaluation metrics are provided in the main paper and Appendices A–E. We make our anonymized source code available at `https://anonymous.4open.science/r/IOHFuseLM-C5A4`. We also provide the preprocessed VitalDB and MIT-BIH Arrhythmia datasets in the supplementary materials.

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

# A DATASET DETAILS

## A.1 INTRAOPERATIVE HYPOTENSION PREDICTION DATASET

Table 5: Statistics of the VitalDB and Clinical IOH datasets under varying sampling settings.

| Dataset | Sampling (s) | Historical Window | Predicted Window | Train | Val | Test | Surgeries | N | IOH$_{train}$ |
|---|---|---|---|---|---|---|---|---|---|
| Clinic-IOH | 6 | 150 | 50 | 2749 | 852 | 884 | 28 | 1452 | 135 |
| | | | 100 | 3348 | 937 | 1138 | | | |
| | | | 150 | 4507 | 1578 | 1804 | | | |
| | 10 | 90 | 30 | 10765 | 3236 | 3481 | | | |
| | | | 60 | 12334 | 3752 | 4227 | | | |
| | | | 90 | 13320 | 4166 | 4609 | | | |
| VitalDB | 3 | 300 | 100 | 18172 | 2308 | 2299 | 12 | 1522 | 2026 |
| | | | 200 | 27031 | 3635 | 3356 | | | |
| | | | 300 | 38479 | 5201 | 4647 | | | |

Table 5 summarizes the key statistics of the two intraoperative hypotension (IOH) datasets used in this study: Clinical IOH and VitalDB. For each dataset, we list the sampling frequency, historical length $l$, predicted horizon $t$, and the number of training, validation, and test samples. The table also includes the number of unique patients ($N$) and surgeries, along with the number of IOH events identified in the training set (IOH$_{train}$) according to our clinical threshold definition. **Clinical IOH Dataset.** This dataset consists of intraoperative data collected from 6,822 patients undergoing anesthesia. It contains high-resolution arterial blood pressure (ABP) waveforms sampled at 100 Hz and structured patient information including age, gender, and surgery type. To accommodate different temporal resolutions, ABP waveforms were processed into MAP series and resampled at 6,s and 10,s intervals. Segments shorter than 1,000 seconds were discarded, yielding 1,452 valid recordings for analysis. Data acquisition was approved by the institutional ethics committee. **VitalDB Dataset.** Derived from the public VitalDB repository, this dataset initially included 6,388 intraoperative recordings with ABP values sampled every 3 seconds. We excluded recordings with more than 20% missing data in the observation window. After filtering, 1,522 high-quality samples remained for downstream tasks. **Data Splitting and Forecasting Settings.** Both datasets are split into training, validation, and test subsets using a 3:1:1 ratio, preserving temporal consistency without shuffling. Each model ingests a fixed 15-minute historical MAP window and predicts MAP trajectories over future horizons of 5, 10, or 15 minutes. These predicted lengths are chosen based on prior clinical research demonstrating their practical relevance for IOH risk forecasting (Awad et al., 2022). The Clinical IOH dataset was de-identified under the HIPAA Safe Harbor method by removing all 18 identifiers. The VitalDB dataset is publicly available under the CC BY-NC-SA 4.0 license, allowing unrestricted use for research. All physiological time-series data are processed exclusively within locally trained models such as GPT-2 and LLaMA, without any exposure to external services. Physiological time series were not transmitted through any external services and all processing was conducted locally.

## A.2 ABNORMAL HEARTBEAT DETECTION DATASET

Table 6: Statistics of the MIT-BIH Arrhythmia dataset.

| Metric | Train | | | Validation | | | Test | | |
|---|---|---|---|---|---|---|---|---|---|
| | Sampling Points | Total Instances | Abnormal | Sampling Points | Total Instances | Abnormal | Sampling Points | Total Instances | Abnormal |
| MIT-BIH Arrhythmia | 15,349,248 | 14,977 | 4,473 | 6,769,664 | 6,608 | 2,650 | 5,900,544 | 5,759 | 1,410 |

As shown in Table 6, the **MIT-BIH Arrhythmia dataset** consists of 48 dual-lead ECG recordings, each lasting approximately 30 minutes, collected from 47 subjects. The signals were sampled at 360 Hz with 11-bit resolution and an amplitude range of $\pm 10$ mV. In total, more than 110,000 heartbeats were manually annotated by clinical experts, covering categories such as normal beats, ventricular beats, fusion beats, and other arrhythmias. **Data Splitting and Forecasting Settings.** In the preprocessing stage, we extracted fixed-length heartbeat segments centered on R-peaks, where each segment contained 128 samples before the peak and 128 samples after the peak, resulting in 256 samples per beat for both ECG leads. Heartbeats annotated as normal (N) were labeled 0, and all arrhythmic types were labeled 1, with segments containing invalid annotations or out-of-range indices excluded. The dataset was divided into training, validation, and test sets in a 28:10:10 ratio.

Input sequences were built with a sliding window of three consecutive beats followed by one target beat. This task used only sequential ECG signals, providing a controlled setting to evaluate the effectiveness of the augmentation, pretraining, and fine-tuning procedures.

## B EXPERIMENT DETAILS

Table 7: Hyperparameter configurations across different datasets and settings.

| Dataset | Sampling (s) | Predicted Window | Pretrain LR | Finetune LR | H | E | $\Delta_{\text{Normal}}$ | $\Delta_{\text{IOH}}$ |
|---|---|---|---|---|---|---|---|---|
| CH-OBPB | 6 | 50 | $10^{-4}$ | $10^{-5}$ | 4 | 5 | 150 | 2 |
| | | 100 | $10^{-5}$ | $5 \times 10^{-5}$ | 4 | 5 | | |
| | | 150 | $10^{-4}$ | $10^{-4}$ | 3 | 2 | | |
| CH-OBPB | 10 | 30 | $10^{-4}$ | $10^{-4}$ | 5 | 2 | 20 | 1 |
| | | 60 | $10^{-5}$ | $5 \times 10^{-5}$ | 4 | 4 | | |
| | | 90 | $5 \times 10^{-5}$ | $10^{-4}$ | 1 | 5 | | |
| VitalDB | 3 | 100 | $3 \times 10^{-5}$ | $10^{-4}$ | 3 | 2 | 150 | 10 |
| | | 200 | $5 \times 10^{-5}$ | $10^{-4}$ | 4 | 2 | | |
| | | 300 | $10^{-5}$ | $10^{-4}$ | 5 | 2 | | |

Table 7 summarizes the hyperparameter configurations used across different datasets and experimental settings. Specifically, it includes the predicted window length $t$, learning rates for domain adaptive pretraining and task fine-tuning, the number of augmented series $H$, the GPT Layers $E$, a hyperparameter denoting the count of stacked Transformer encoder layers in the GPT-2 architecture adopted by our framework, as well as the sampled intervals $\Delta_{\text{Normal}}$ and $\Delta_{\text{IOH}}$. Our framework is based on GPT-2. The batch size is fixed at 4 for pretraining and 8 for fine-tuning. The pretraining masking ratio $R$ is set to 0.2, and the hyperparameter $\rho$ of IOH loss is set to 10. The diffusion process utilizes $K = 50$ steps with a cosine variance schedule (Rasul et al., 2021) from $\beta_1 = 10^{-4}$ to $\beta_K = 0.5$. Most baseline models and our proposed method are evaluated on a NVIDIA RTX 4090 GPU to ensure a fair comparison under practical deployment settings. However, due to substantial memory requirements of TimeLLM, which is based on the LLaMA 7B language model, both training and inference for this model are conducted on a server equipped with NVIDIA A100 GPUs.

## C DESIGN OF MULTI-SCALE TREND-RESIDUAL DIFFUSION AUGMENTATION

As shown in the Figure5, the diffusion augmentation network $f_\theta$ is implemented as a lightweight denoising module for residual series refinement. Given the noisy residual input $x_{i,\text{residual}}^{(k)}$, the network first projects it into a latent space using a multilayer perceptron (MLP) with hidden size $d$, where positional information is injected through learnable embeddings and sinusoidal encodings of the diffusion step $k$, fused via Adaptive Layer Normalization (AdaLN). The latent features are then processed by a stack of $N$ decoder blocks, each combining a linear projection, AdaLN, and a feed-forward sublayer with residual connections. This structure enables stable training and efficient denoising while avoiding the computational overhead of heavy convolutional or attention-based modules. The final linear projector maps the refined representation back to the residual space, yielding $\hat{x}_{i,\text{residual}}^{(0)}$, which is combined with the multi-scale trend to reconstruct the augmented MAP series. This streamlined design effectively captures fine-grained fluctuations in the residual series while maintaining computational efficiency, enabling the generation of realistic IOH augmentations.

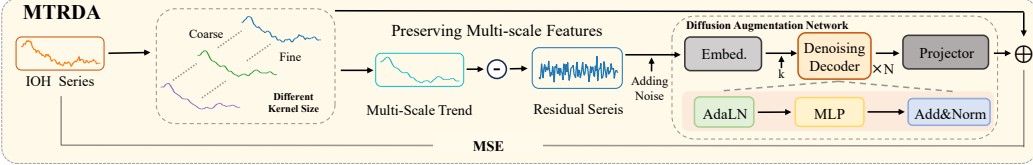

Figure 5: Overview of the proposed Multi-Scale Trend-Residual Diffusion Augmentation (MTRDA).

## D PROMPT FOR PERSONALIZED CLINICAL DESCRIPTION GENERATION

To generate patient-specific clinical narratives, we design a structured prompt that guides GPT-4o to produce medically grounded descriptions. The prompt integrates static attributes such as age, gender, and surgery type with domain-informed templates, as illustrated in Figure 6. The resulting text provides a personalized semantic representation for multimodal fusion in the forecasting pipeline.

**Prompt Template:**

```
The age of patient is {age of
patient}, gender is {gender
of patient}, and the type of
surgery is {surgery type of
patient}.  Please provide the
answer directly, separated by
commas, without any spaces in
between, removing the parentheses
when responding.  Without any
explanations or additional content.
The patient belongs to the () age
group, whose vascular compliance
and cardiovascular compensatory
capacity are ().  At this time,
() hormones act on the blood
vessels.  This surgery is a ()
type of surgery, and the blood
loss is usually ().
```

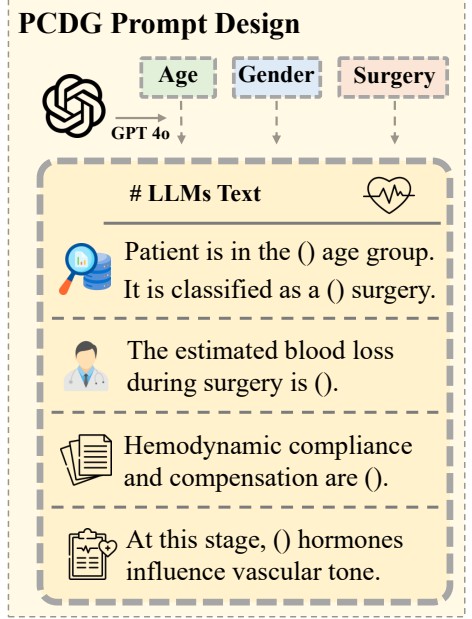

Figure 6: Illustration of the PCDG (Personalized Clinical Description Generation) prompt design. The framework integrates static attributes into structured templates, which GPT-4o converts into concise clinical narratives.

This prompt in Personalized Clinical Description Generation (PCDG) balances consistency with clinical variability by incorporating patient-specific attributes. It is tokenized with an extended vocabulary covering physiological and surgical terms, enabling the model to embed static medical context into prediction.

## E HYPERPARAMETER SENSITIVITY

Table 8: Performance of the model under different historical and predicted lengths.

| $t$ | $l = 30$ | | | | $l = 60$ | | | | $l = 90$ | | | |
|---|---|---|---|---|---|---|---|---|---|---|---|---|
| | $\text{MSE}_{\text{IOH}}$ | $\text{MAE}_{\text{IOH}}$ | Recall | AUC | $\text{MSE}_{\text{IOH}}$ | $\text{MAE}_{\text{IOH}}$ | Recall | AUC | $\text{MSE}_{\text{IOH}}$ | $\text{MAE}_{\text{IOH}}$ | Recall | AUC |
| 30 | $120.85 \pm 11.18$ | $8.98 \pm 0.67$ | 0.482 | 0.6852 | $101.09 \pm 9.34$ | $8.28 \pm 0.64$ | 0.5743 | 0.7288 | $43.29 \pm 8.64$ | $4.65 \pm 0.20$ | 0.7205 | 0.7729 |
| 60 | $72.73 \pm 13.06$ | $6.75 \pm 0.90$ | 0.8191 | 0.7596 | $65.70 \pm 12.91$ | $6.03 \pm 0.82$ | 0.7822 | 0.7444 | $82.07 \pm 11.10$ | $6.88 \pm 0.77$ | 0.8012 | 0.7693 |
| 90 | $56.93 \pm 5.44$ | $5.88 \pm 0.48$ | 0.8368 | 0.7184 | $85.45 \pm 2.79$ | $7.45 \pm 0.45$ | 0.7619 | 0.7318 | $91.16 \pm 15.48$ | $7.25 \pm 0.69$ | 0.7976 | 0.7642 |

To assess the sensitivity of the model to varying historical and predicted horizons, we evaluate performance across different combinations of historical length $l$ and predicted length $t$. As shown in Table 8, increasing the historical length generally improves performance across all metrics. The setting with $l = 90$ and $t = 30$ achieves the best overall results, with the lowest prediction error, indicating that a longer temporal context enhances short-term IOH risk forecasting. In contrast, extending the predicted length leads to a moderate decline in accuracy, reflecting the increased difficulty of long-range forecasting in clinical settings.

We conduct a sensitivity analysis on the Clinic IOH dataset under 10-second sampling resolution to evaluate the impact of key hyperparameters. As shown in Figure 7, the results indicate that model performance is moderately sensitive to the fine-tuning learning rate, while the pretraining learning rate exhibits greater stability. Varying the number of GPT layers shows that moderate depth achieves better generalization, whereas excessive depth may lead to overfitting. Additionally, appropriate levels of data augmented by MTRDA consistently improve performance, though excessive augmentation

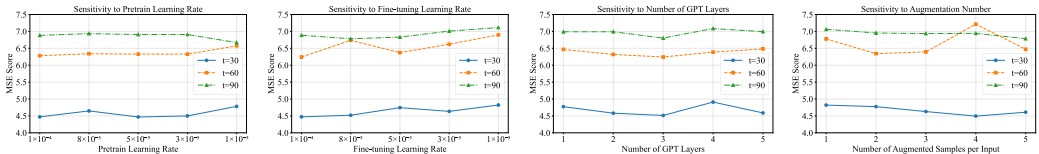

Figure 7: Parameter sensitivity analysis on the Clinic IOH dataset.

can introduce distributional noise and degrade accuracy. These findings highlight the importance of balanced model capacity and augmentation strategies for stable performance.

# F    VISUALIZATION

Figure 8: Visual comparison of MAP prediction results across different models.

### F.1    MODEL PREDICTION VISUALIZATION

Figure 8 presents a visual comparison of seven models under the 6-second sampling granularity, with a historical window length $l = 150$ and a predicted horizon $t = 150$. Each row corresponds to one model and each column represents a distinct IOH case. It can be observed that our proposed model consistently identifies hypotensive risks across all three representative IOH events, demonstrating both precise forecasting accuracy and effective event discrimination. In contrast, among the baseline models, only Fredformer correctly identifies the third IOH event, while the others fail to capture the hypotensive onset in this scenario.

### F.2    AUGMENTATION VISUALIZATION

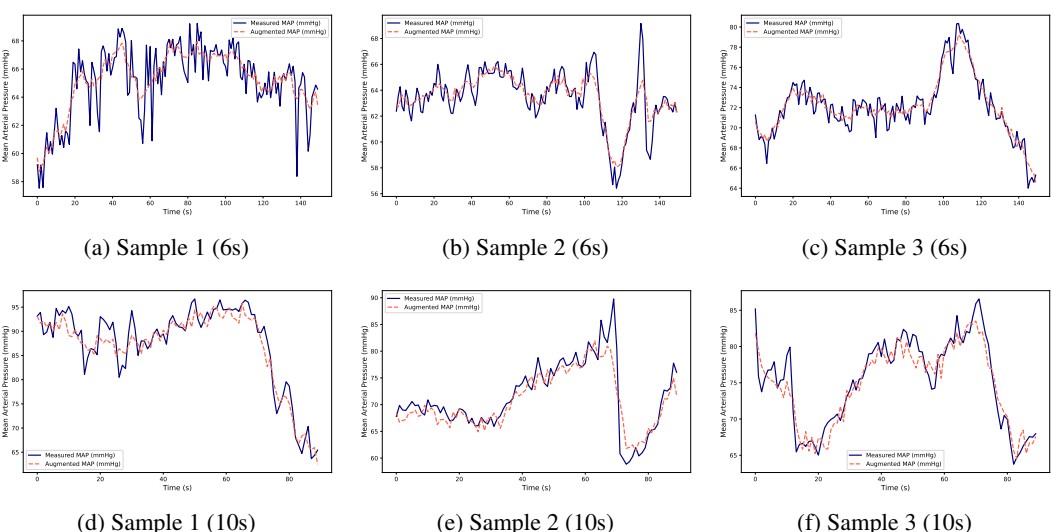

(a) Sample 1 (6s)   (b) Sample 2 (6s)   (c) Sample 3 (6s)

(d) Sample 1 (10s)   (e) Sample 2 (10s)   (f) Sample 3 (10s)

Figure 9: Examples of augmented MAP series of MTRDA under different sampling frequencies.

Figure 9 presents representative examples of MAP time series augmented by the proposed MTRDA framework under two different sampling frequencies. The augmented series preserve the trends extracted through multiscale smoothing and simultaneously introduce fine-grained variations that enrich the temporal structure of the original series. In particular, the augmented outputs retain the essential characteristics of hypotensive episodes while reducing noise, reflecting the ability of MTRDA to reconstruct physiologically meaningful patterns through trend-residual decomposition and diffusion-based enhancement. These results confirm the effectiveness of MTRDA in improving the representation quality of sparse IOH series under varying temporal resolutions.

## G    ANALYSIS UNDER HETEROGENEOUS SAMPLING FREQUENCIES

In real-world clinical practice, different hospitals and monitoring devices often adopt heterogeneous sampling frequencies, resulting in varied temporal resolutions of intraoperative arterial pressure recordings. To examine the adaptability of IOHFuseLM to such practical discrepancies, we conducted additional experiments under both finer and coarser sampling settings, as well as transfer evaluations simulating deployment in low-resolution environments.

### G.1    PERFORMANCE UNDER FINER AND COARSER SAMPLING RATES

We first evaluated the Clinical IOH dataset with alternative sampling granularities of 3 seconds and 12 seconds. As shown in Table 9, IOHFuseLM consistently outperforms all baselines, demonstrating strong adaptability to both fine-grained and coarse-grained temporal resolutions.

Table 9: Performance comparison under different sampling resolutions.

| Model | 3-second | | | | 12-second | | | |
|---|---|---|---|---|---|---|---|---|
| | $MSE_{IOH}$ | $MAE_{IOH}$ | Recall (%) | AUC | $MSE_{IOH}$ | $MAE_{IOH}$ | Recall (%) | AUC |
| DLinear | 154.70 | 9.29 | 52.62 | 0.6938 | 187.13 | 11.51 | 17.18 | 0.5409 |
| PatchTST | 107.18 | 7.82 | 66.29 | 0.7093 | 168.06 | 10.57 | 30.55 | 0.5650 |
| Fredformer | 105.40 | 7.86 | 55.92 | 0.6985 | 151.31 | 10.18 | 22.32 | 0.5443 |
| HMF | 116.52 | 7.93 | 61.47 | 0.7173 | 180.86 | 11.40 | 13.40 | 0.5235 |
| GPT4TS | 115.32 | 8.20 | 59.42 | 0.7064 | 159.68 | 10.36 | 26.62 | 0.5642 |
| **IOHFuseLM** | **103.69** | **7.39** | **70.12** | **0.7251** | **136.22** | **9.57** | **36.14** | **0.5730** |

## G.2 LOW-RESOLUTION TRANSFER EVALUATION

To emulate clinical scenarios with sparse or low-quality monitoring, we designed a low-resolution transfer evaluation. Specifically, each 12-second sampling point was duplicated to emulate a 6-second sampling interval, thereby reducing information density while preserving sequence length. We considered two experimental settings: (i)Transfer: training on high-fidelity data and testing on degraded low-resolution inputs; (ii)Non-transfer: both training and testing on low-resolution inputs. Results in Table 10 show that IOHFuseLM demonstrates strong robustness under both scenarios, with particularly notable generalization in the transfer setting, achieving the highest recall and AUC among all baselines.

Table 10: Performance comparison under the Low-Resolution Transfer Evaluation.

| Model | Transfer | | | | Non-transfer | | | |
|---|---|---|---|---|---|---|---|---|
| | $MSE_{IOH}$ | $MAE_{IOH}$ | Recall (%) | AUC | $MSE_{IOH}$ | $MAE_{IOH}$ | Recall (%) | AUC |
| DLinear | 161.85 | 10.20 | 39.30 | 0.6019 | 194.69 | 11.84 | 22.95 | 0.5573 |
| PatchTST | 120.92 | 8.81 | 52.87 | 0.6329 | 170.04 | 10.70 | 44.04 | 0.6061 |
| Fredformer | 135.67 | 9.60 | 38.68 | 0.6026 | 152.91 | 10.28 | 33.69 | 0.5862 |
| HMF | 135.81 | 9.57 | 41.00 | 0.6096 | 163.14 | 10.75 | 23.66 | 0.5529 |
| GPT4TS | 113.12 | 8.44 | 50.23 | 0.6370 | 159.35 | 10.35 | 41.51 | 0.6143 |
| **IOHFuseLM** | **106.54** | **6.84** | **76.76** | **0.6637** | **136.52** | **9.34** | **48.95** | **0.5985** |

These results confirm that IOHFuseLM maintains superior performance under heterogeneous sampling conditions, highlighting its adaptability to diverse clinical environments. Importantly, the transfer evaluation demonstrates that training on high-fidelity data enables effective generalization to degraded inputs, underscoring the practical utility for deployment across hospitals with varying acquisition protocols.

## H PRACTICAL DEPLOYMENT AND BROADER IMPACTS

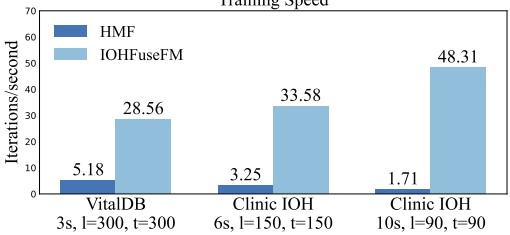 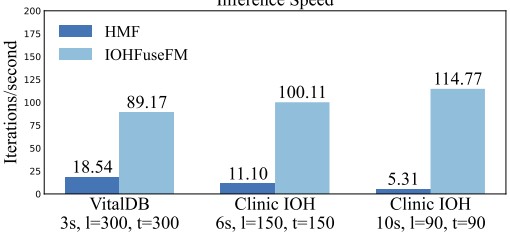

Figure 10: Training and inference efficiency comparison between IOHFuseLM and HMF.

By complementing physiological time series with structured patient descriptions, our framework enables more personalized and context-aware clinical event recognition in settings where signal-only models are often insufficient. This multimodal design positions IOHFuseLM as a promising blueprint for developing robust and adaptive monitoring systems that are more closely aligned

with patient-specific risk profiles. To further evaluate its practical deployability, we compared the computational efficiency of IOHFuseLM with HMF (Cheng et al., 2024), a representative baseline for IOH prediction. As shown in Figure 10, IOHFuseLM consistently achieves faster training and inference across different configurations on both the VitalDB and Clinical IOH datasets. These results demonstrate not only its strong modeling capability but also its computational efficiency, suggesting that the framework can be integrated into real-time clinical workflows. In time-critical environments such as intraoperative monitoring, such efficiency gains have the potential to translate into more reliable and actionable clinical decision support.

Beyond computational efficiency, IOHFuseLM has been developed with practical deployment in mind. Its input requirements are minimal, relying on real-time MAP series derived from arterial blood pressure and structured patient attributes that are already routinely collected by anesthesia monitors, enabling seamless integration with existing perioperative information systems. In addition, the model exhibits stable performance across different sampling rates and levels of event sparsity, underscoring its robustness under heterogeneous monitoring conditions. Its relatively low memory footprint and short inference latency further enhance its suitability for deployment on standard hospital hardware, without the need for specialized accelerators. Finally, while the present study focuses on IOH prediction, the same framework has also demonstrated transferability to related tasks such as arrhythmia detection, indicating its potential extensibility to a broader range of critical care monitoring applications.

# I DERIVATION OF MULTI-SCALE TREND–RESIDUAL DIFFUSION AUGMENTATION

We provide a rigorous derivation for the Multi-Scale Trend–Residual Diffusion Augmentation (MTRDA) module in a single narrative.

We begin with the decomposition of the historical MAP series $\boldsymbol{x}_{i,1:l} \in \mathbb{R}^l$. For each odd-length window $w_s \in \mathcal{S}$, a centered moving-average operator $\boldsymbol{S}_{w_s}$ with symmetric padding is defined. The multiscale trend estimate is expressed as

$$\boldsymbol{x}_{i,\text{trend}} = \frac{1}{|\mathcal{S}|} \sum_{s=1}^{|\mathcal{S}|} \boldsymbol{S}_{w_s} \boldsymbol{x}_{i,1:l}, \tag{13}$$

while the residual is obtained as

$$\boldsymbol{x}_{i,\text{residual}} = \boldsymbol{x}_{i,1:l} - \boldsymbol{x}_{i,\text{trend}}. \tag{14}$$

Equivalently, letting $\bar{\boldsymbol{S}} = \frac{1}{|\mathcal{S}|} \sum_s \boldsymbol{S}_{w_s}$, we have $\boldsymbol{x}_{i,\text{trend}} = \bar{\boldsymbol{S}} \boldsymbol{x}_{i,1:l}$ and $\boldsymbol{x}_{i,\text{residual}} = (\boldsymbol{I} - \bar{\boldsymbol{S}})\boldsymbol{x}_{i,1:l}$. Since $\bar{\boldsymbol{S}}$ is symmetric and row-stochastic, the residual subspace captures high-frequency fluctuations while the trend subspace preserves the baseline.

Defining $\boldsymbol{r}_i^{(0)} := \boldsymbol{x}_{i,\text{residual}}$, the forward diffusion process follows the DDPM formulation:

$$q(\boldsymbol{r}_i^{(k)} \mid \boldsymbol{r}_i^{(k-1)}) = \mathcal{N}\left(\sqrt{\alpha_k}\boldsymbol{r}_i^{(k-1)}, \beta_k \boldsymbol{I}\right), \tag{15}$$

$$q(\boldsymbol{r}_i^{(k)} \mid \boldsymbol{r}_i^{(0)}) = \mathcal{N}\left(\sqrt{\bar{\alpha}_k}\boldsymbol{r}_i^{(0)}, (1 - \bar{\alpha}_k)\boldsymbol{I}\right), \tag{16}$$

with $\alpha_k = 1 - \beta_k$ and $\bar{\alpha}_k = \prod_{j=1}^k \alpha_j$. This implies the reparameterization

$$\boldsymbol{r}_i^{(k)} = \sqrt{\bar{\alpha}_k}\,\boldsymbol{r}_i^{(0)} + \sqrt{1 - \bar{\alpha}_k}\,\boldsymbol{\epsilon}, \quad \boldsymbol{\epsilon} \sim \mathcal{N}(\boldsymbol{0}, \boldsymbol{I}). \tag{17}$$

The exact reverse posterior given $\boldsymbol{r}_i^{(0)}$ is Gaussian:

$$q(\boldsymbol{r}_i^{(k-1)} \mid \boldsymbol{r}_i^{(k)}, \boldsymbol{r}_i^{(0)}) = \mathcal{N}\left(\boldsymbol{m}_k(\boldsymbol{r}_i^{(0)}, \boldsymbol{r}_i^{(k)}), \tilde{\beta}_k \boldsymbol{I}\right), \tag{18}$$

where the mean is

$$\boldsymbol{m}_k(\boldsymbol{r}^{(0)}, \boldsymbol{r}^{(k)}) = \frac{\sqrt{\bar{\alpha}_{k-1}}\beta_k}{1 - \bar{\alpha}_k}\,\boldsymbol{r}^{(0)} + \frac{\sqrt{\alpha_k}(1 - \bar{\alpha}_{k-1})}{1 - \bar{\alpha}_k}\,\boldsymbol{r}^{(k)}. \tag{19}$$

The parametric reverse process in MTRDA is defined as

$$p_\theta(\boldsymbol{r}_i^{(k-1)} \mid \boldsymbol{r}_i^{(k)}, \boldsymbol{c}_i) = \mathcal{N}\big(\boldsymbol{\mu}_\theta(\boldsymbol{r}_i^{(k)}, k, \boldsymbol{c}_i), \sigma_k^2 \boldsymbol{I}\big), \tag{20}$$

where $\boldsymbol{c}_i$ includes conditioning on the trend component and local patch statistics.

The conditional ELBO for $\log p_\theta(\boldsymbol{r}_i^{(0)} \mid \boldsymbol{c}_i)$ decomposes into KL terms:

$$\mathcal{L}_{\text{ELBO}} = -\sum_{k=1}^{K} \mathbb{E}_q \Big[ D_{\text{KL}} \Big( q(\boldsymbol{r}_i^{(k-1)} \mid \boldsymbol{r}_i^{(k)}, \boldsymbol{r}_i^{(0)}) \parallel p_\theta(\boldsymbol{r}_i^{(k-1)} \mid \boldsymbol{r}_i^{(k)}, \boldsymbol{c}_i) \Big) \Big] + \text{const.} \tag{21}$$

Choosing $\sigma_k^2 = \tilde{\beta}_k$ reduces each KL term to an MSE between posterior and model means:

$$\mathcal{L}_{\text{ELBO}} \equiv -\sum_{k=1}^{K} \frac{1}{2\sigma_k^2} \mathbb{E}_q \Big\| \boldsymbol{m}_k(\boldsymbol{r}_i^{(0)}, \boldsymbol{r}_i^{(k)}) - \boldsymbol{\mu}_\theta(\boldsymbol{r}_i^{(k)}, k, \boldsymbol{c}_i) \Big\|_2^2 + \text{const.} \tag{22}$$

Adopting the $x_0$-prediction parameterization, we let

$$\boldsymbol{\mu}_\theta(\boldsymbol{r}^{(k)}, k, \boldsymbol{c}) = a_k \boldsymbol{r}^{(k)} + b_k \, \boldsymbol{r}_\theta(\boldsymbol{r}^{(k)}, k, \boldsymbol{c}), \tag{23}$$

where

$$a_k = \frac{\sqrt{\alpha_k}(1 - \bar{\alpha}_{k-1})}{1 - \bar{\alpha}_k}, \qquad b_k = \frac{\sqrt{\bar{\alpha}_{k-1}} \beta_k}{1 - \bar{\alpha}_k}.$$

The ELBO then reduces to

$$\min_\theta \sum_{k=1}^{K} w_k \, \mathbb{E}_{\boldsymbol{r}^{(0)}, k, \boldsymbol{\epsilon}} \Big\| \boldsymbol{r}^{(0)} - \boldsymbol{r}_\theta(\boldsymbol{r}^{(k)}, k, \boldsymbol{c}) \Big\|_2^2, \quad \boldsymbol{r}^{(k)} = \sqrt{\bar{\alpha}_k} \boldsymbol{r}^{(0)} + \sqrt{1 - \bar{\alpha}_k} \, \boldsymbol{\epsilon}, \tag{24}$$

which matches the simplified denoising loss described in the main text.

Finally, the augmented sequence is reconstructed as

$$\hat{\boldsymbol{x}}_{i,1:l} = \boldsymbol{x}_{i,\text{trend}} + \hat{\boldsymbol{r}}_i^{(0)}, \tag{25}$$

where $\hat{\boldsymbol{r}}_i^{(0)}$ is sampled from the trained denoising network $f_\theta$. In expectation, the reconstruction preserves the baseline trend while enriching the residual component with fine-grained fluctuations, ensuring that the generated MAP series capture both stability and informative variability.

## J  THE USE OF LARGE LANGUAGE MODELS

Large language models were used as auxiliary tools for paper polishing, primarily to improve fluency and readability. Their role was limited to linguistic refinement and did not affect the experimental results or conclusions.

