# OpenReview forum: "Multimodal Forecasting of Sparse Intraoperative Hypotension Events Powered by Language Model"
_ICLR.cc/2026/Conference — ICLR 2026 Conference Withdrawn Submission_

### Official Review · Reviewer_Mig5 · 2025-11-01

**Soundness:** 3
**Presentation:** 4
**Contribution:** 3
**Rating:** 6
**Confidence:** 3

**Summary:**

This paper proposes IOHFuseLM, a novel multimodal language model framework for forecasting sparse intraoperative hypotension (IOH) events, a clinically significant and challenging problem. The core contribution lies in an innovative strategy to address the dual challenges of heterogeneous data fusion and event sparsity.
1.The PCDG module , which uses a large language model (GPT-40) to "translate" static patient attributes (age, gender, surgery type) into rich semantic text descriptions.
2.A token-level cross-attention mechanism to fuse these text descriptions with dynamic MAP time-series patch embeddings.
3.The MTRDA module , a diffusion-based augmentation strategy based on trend-residual decomposition, specifically designed to generate realistic, sparse IOH signals.
4.A two-stage paradigm of domain-adaptive pretraining (on augmented data) and task-finetuning (with an IOH-weighted loss) .

**Strengths:**

1.Important and Well-Defined Problem: The paper targets IOH prediction, a significant clinical challenge . It accurately identifies the core bottlenecks of existing methods: event sparsity and heterogeneous data fusion.
2.High Novelty (PCDG): The PCDG module  is an outstanding innovation. It creatively uses a powerful LLM (GPT-4o) as an advanced feature encoder to "textify" static attributes. This allows them to be elegantly fused with time-series patch embeddings via token-level cross-attention, which is far more sophisticated than traditional feature concatenation.
3.High Reproducibility: The paper provides an anonymized code link and critical reproduction details in the appendix, including hyperparameters (App B ), MTRDA architecture (App C ), and the exact PCDG prompts (App D ).

**Weaknesses:**

1.Dependency on GPT-4o: The framework's first critical step, PCDG , relies on a closed-source, expensive, and potentially non-stationary LLM (GPT-4o). This significantly harms reproducibility, increases deployment costs, and introduces an external dependency. Suggestion: The authors should add an ablation using a smaller, open-source model (e.g., LLaMA-3-8B or a medical-domain-specific model) to perform PCDG and report the performance difference.
2.Unvalidated Assumption in MTRDA: The MTRDA module  is built on the strong assumption that IOH predictive signals reside primarily in the residual component. This is not empirically proven. For example, a slow, steady decline due to blood loss might be captured more in the trend. The authors should provide empirical evidence (e.g., spectral analysis of IOH vs. normal segments) to support this core assumption or discuss cases where it might not hold.
3.Inappropriate Efficiency Comparison: The efficiency analysis in Appendix H (Fig. 10) compares IOHFuseLM to HMF, a non-LLM baseline. A more meaningful comparison would be against GPT4TS (also GPT-2 based). The overall training cost of IOHFuseLM, which includes multi-stage pre-processing (PCDG, MTRDA) and two-stage training, is likely much higher than the baselines. A total end-to-end training time comparison is needed.
4.Typo: The paper repeatedly uses "GPT-40" or "Gpt-40", which appears to be a typo for "GPT-4o".

**Questions:**

1.Robustness of PCDG: Can you quantify the robustness of the PCDG module? Specifically, what is the performance drop (in Recall and AUC) if a smaller, open-source model (e.g., LLaMA-3-8B) is used to generate the clinical descriptions instead of GPT-4o?
2.Direction of Multimodal Fusion: In Sec 4.3, you use the text tokens (PCDG) as the Query and the time-series patches as the Key/Value in your cross-attention36. Did you experiment with the reverse (patches as Query, text as K/V) or a bidirectional fusion mechanism? What was the motivation for this specific design?
3.Impact of 'Surgery-Aware' Splitting: Your "Surgery Aware Subject Splitting" is a strong evaluation protocol. Does this result show that the model performs worse on the unseen surgery types in the test set compared to seen types? In other words, has the model learned a generalizable relationship between types of procedures and IOH patterns, or just memorized the patterns for the specific surgeries seen in training?

**Details Of Ethics Concerns:**

The datasets used are either de-identified (Clinical IOH ) or public benchmarks (VitalDB, MIT-BIH ), adhering to ethical guidelines. The authors provide a clear Ethics Statement in Sec 7

---

### Official Review · Reviewer_a34K · 2025-11-03

**Soundness:** 2
**Presentation:** 2
**Contribution:** 2
**Rating:** 2
**Confidence:** 4

**Summary:**

This paper tackles the problem of detecting intraoperative hypotensive (IOH) events.

The basic architecture is to consume two kinds of input:

* univariate measurements of mean arterial pressure (MAP) over time
* static patient features (not defined initially but presumably facts like age/gender/surgery type, given later text in Sec 5).

The idea is that the MAP time series and static info can be embedded as tokens and processed by a "pretrained language model", followed by a projection linear layer to make predictions.

The training occurs in two phases:

* The first step involves domain-adaptive pretraining, using augmented data from a custom diffusion module. The idea is to learn "semantically meaningful interactions" between the IOH related MAP vital sign time series and corresponding static patient features.
* The second step involves fine-tuning on the targeted real clinical dataset (no augmentation), to refine a classifier of normal vs hypotensive events.

Experiments look at the ability to predict IOH several minutes ahead of time, using two intraoperative datasets where MAP signals and thus IOH ground truth can be found: ClinicalIOH and VitalDB. Experiments also assess the ability to predict abnormal heart beats in the MIT/BI Arrhythmia dataset.

**Strengths:**

* The focus on a real clinical problem (IOH) is interesting. I do appreciate the task-first rather than method-first approach.
* Experiments in Table 1 compare to sufficient alternative methods that cover both other general purpose approaches and recent methods specific to IOH
* The attention to ablations in Sec. 5 is welcome and seems reasonably thorough
* The approach seems sufficiently original, in that I haven't seen token-level "alignment" of vital sign embeddings and descriptions of static patient factors like age/sex/surgery type

**Weaknesses:**

Overall, this is an interesting approach but I worry about the soundness, clarity and reproducibility of the methods description (see concerns M1-M5 below) as well as the thoroughness and reproducibility of the experimental design (see concerns E1-E8).


## Issues with modeling approach

### M1: Value of the prediction task in real clinical workflow remains unclear

The model is intended to predict intraoperative hypotension (IOH) events, when

> mean arterial pressure (MAP) remains below 65 mmHg for at least one continuous minute

But as visualized in Fig 1, it seems the MAP is measured at high frequency and IOH events can easily be detected from raw sensor data (when value drops below 65). So as soon as this value has dropped below the critical value for more than X seconds, an alert could be made just by monitoring the raw blood pressure readings.

So the added value of a forecasting system must be to detect when this might happen ahead of time. But how far ahead? And what can you do about it?

Fig 2 itself suggests the "historical" input signal to the prediction starts about 200 seconds before the MAP event and ends about 50 seconds before the event. What is the value of an alert that happens 50 seconds before the event will be easily readable on a bedside sensor? What interventions are possible and what is the added value of an extra 50 sec of notice?

Later in Sec 5, the paper seems to use "prediction horizons of 5, 10, and 15 minutes". So I guess counter to what's shown in Fig 2, the model tries to predict 5 or more minutes ahead. This seems more actionable, but again, what would you do about it? What's the intervention that's possible, and is 5 minutes warning enough to do it? Beyond just recall, other metrics like true negative rate and positive predictive value are probably necessary to judge if model-supported interventions are worthwhile.


### M2: Value of the diffusion model seems constrained since trend signal not modified

The diffusion model is set up to only perturb the residuals, not impact the core trend (obtained by weighted averages of the raw signal at multiple scales). If the core trend is not perturbable, I don't see how this can generate lots of interesting other plausible MAP signals with or without IOH.

For example, I don't think this procedure can:

* add shifts or delays in the core trend signal
* change the amplitude of the core trend signal in consistent fashion

Can the authors comment on why flexible models of residuals alone are enough to achieve interesting variability in the sampled time series?


### M3: Does the intended value of the domain adaptive pretraining have a clinical basis?

The domain-adaptive efforts in Sec. 4.3 seem abstractly interesting, but can we really explain most of the variations in MAP signals by age or sex or other static factors? I'd expect background demographics or surgery type to explain very little of the interesting signal in MAP. Is there documented clinical evidence that such factors play a substantial role in MAP patches? E.g. are their well known patterns in MAP signals that clinicians refer to as being particularly associated with the elderly or with certain types of surgeries?

### M4: Loss design for the fine-tuning does not seem understandable or reproducible

I don't understand Eq 12 or know how to implement Eq 12. The dimensionality of the terms in the RHS is unclear... what is used to compute the MSE values there?

* Is Eq 12 saying that each *timestep* in a MAP series can be labeled as either IOH or not, and we weight the IOH timesteps more than other timesteps?
* Or is Eq 12 saying that sequences labeled normal (no IOH events within them at all) are upweighted?
* Or is it something else? Maybe Eq 12 intends that the predicted start time of IOH matches the truth well?

Also, is the intended value of $\rho$ smaller or larger than one? It is not quite clear as written.

### M5: Diffusion module description in Sec 4.2 needs work to be understandable to broad audience and reproducible

Currently I am not confident another capable researcher with solid background could reproduce the diffusion module based on its provided description.

Usually, I'd expect to see both forward and reverse processes described to define a diffusion approach. I just see one way here. It's unclear how "reconstruction" happens as intended, if you are just adding noise as in Eq 7.

Eq 8 has several issues:
* what is the distribution the expectation is with respect to? This is not defined clearly to me.
* why use the subscript "ELBO"? This looks quite different from the ELBO (evidence lower bound) of variational inference, for example.
* Unclear what symbol $x_{i,residual,k}$ with k in subscript means... k was denoted differently elsewhere.

I'm also confused by the text that says after sampling a residual, it

> combines it with the trend component $x_{i,trend}$ to generate the output and compute the training loss

Is this loss the same as the loss in Eq 8? If so, why do you need to add the trend? Eq 8 only compares to the true residual. If the "training loss" is something other than Eq 8, it needs to be defined clearly.


## Experimental issues

### E1: Description of "surgery-aware splitting" doesn't make sense to me

In lines 164-167, the paper says

> Patients are grouped according to their surgery type, and each group is assigned to only
one data partition. This stratification helps maintain a balanced distribution of surgery types across splits, thus mitigating any distributional shifts caused by surgery specific hypotension risks

I don't see how the claims in this second sentence can be true. If all patients from vascular are assigned to the train set, there won't be any vascular patients in validation or testing, so there's definitely not a "balanced distribution of surgery types across splits", at least interpreting splits as train/validation/test. Instead of "mitigating any distributional shifts", this will ensure there are definitely surgery-type related distributional shifts, as there may be colorectal complications that are unseen in the split not containing colorectal patients.

Perhaps what's intended is that the train/validation/test split is made by stratifying by surgery type? But this is not what's written or what's drawn in Fig 2.

### E2: Description of data insufficient at top of Sec 5

Here's a list of issues related to data description and reproducibility

* (i) Sec 5 says the raw vital sign data is

> MAP time series resampled at 6 and 10 seconds from arterial blood pressure waveforms

What does the "at 6 and 10 seconds" mean here? Presumably there's a regular MAP measurement every X seconds, as depicted in Fig 1. Do you mean "every 6 seconds" or "every 10 seconds"? Presumably you have to pick one or the other.

* (ii) The ClinicalIOH dataset does not have a citation. In the main paper, there's not enough information about where the data comes from (one site? multiple sites?) and what kind of patients are in the data. I guess this might be a custom dataset gathered for this paper, but it is not obvious from the main paper. The paper should be clear about whether this data is available to others, or not.

* (iii) From text of Sec 5, it is not clear for VitalDB what patient attributes are available. The same as ClinicalIOH?

* (iv) The text of Sec 5 mentions several prediction horizons are considered, but it is not clear in Tab 1 or Tab 2 which one horizon (of many possible) is used.

### E3: Implementation architecture not fully described

The presented method is described as using a "pretrained language model" but it is not clear what architecture is used.... Llama? Qwen? something else in the open weight family?

This is a critical reproducibility issue.

### E4: Performance metrics in Table 1 need more clarity

For MSE/MAE, how is this computed? The main paper just says "on hypotensive timestamps".
Does this mean error in the predicted vs actual timestamps where IOH occurs? Or does this mean you look only at timestamps where the true MAP signal falls below the threshold, and compute error between predicted and actual MAP values?

For recall and AUC, is the task a per-timestamp binary classification of IOH or not? How is the thresholding done to compute recall?

### E5: Hyperparameters of baseline models may not be fairly adjusted

It seems from the brief description in lines 340-354, that baselines are each trained using the same fixed learning rate and batch size, which is quite small (only 8 sequences at a time!).

Typically, when baselines are given full consideration with proper hyperparameter tuning, their performance often improves substantially. There still may be good reasons this approach does better, but I'm not sure these experiments give adequate training/tuning effort to the competitor methods.

### E6: Use of averages over multiple runs needs clarity

The paper says around lines 340-354 that

> To ensure result reliability, we report averages over three independent runs

What needs to be clarified here is what varies over the runs, and why an average is appropriate. If you are varying the selecting of the train/valid/test sets, perhaps an average is appropriate.

However, if you are just varying the random initialization of parameters, and then looking the results of stochastic gradient descent, it is not clear to me that an "average" is  the best thing to do in this situation. Reporting an average will get you the method that typically performs well across initializations. However, some methods may get stuck more often in local optima, but their best results may exceed the typical result of alternatives. Such methods would not be rewarded by this process.

I'd hope a revision makes a better case for why reporting typical performance is the preferred method here, since this may not result in the best possible model in terms of ultimate generalization performance.

### E7: Claims about why some methods do better than others do not seem directly supported by data

Several claims in the "Main results" paragraph do not seem directly supported by the provided evidence in Table 1. All that Table 1 can tell us is which methods score best on certain metrics on certain datasets.

Yet the text says things like:

* "Fredformer improves performance by reducing frequency bias but struggles with the high variability of IOH events". There is no evidence presented that it reduces frequency bias (none of the metrics in Table 1 measure this directly). It is also not clear that the "high variability of IOH" is the reason for any lower numbers in Table 1.

Just about every sentence here (not just about Fredformer, but other methods too) makes a rather strong conjecture about why certain methods do better than others that is not obviously supported by the numbers in Table 1. These claims may be true, but I think the narrative here needs some work to acknowledge what claims are just conjectures about why one method works well and which are supported by the evidence in the Tables.

### E8: Abnormal heart beat prediction task needs more justification

I appreciate the effort to look at another task beyond IOH, but I wonder:

* why is the HCM model, which is framed by its creators as specifically for IOH not a general purpose time series predictor, an appropriate baseline here?
* why only compare to Time LLM and not other baselines from Table 1 that seem capable as general time series classifiers?

**Questions:**

## Questions

See my questions throughout M1-M5 and E1-E8 above.

If you need to prioritize something, I think M1, M3, and M4 as well as E1 (justification/clarification of "surgery-aware splitting") and E5 (baselines need to tune hyperparameters) are the most important to my decision at the moment, provided other clarity issues are resolved.

**Details Of Ethics Concerns:**

The paper uses a dataset "ClinicalIOH" that seems to be locally-procured measurements of intraoperative vital signs and outcomes from real surgery patients. I'd want to confirm the IRB/ethics approval of this dataset usage, though the provided Ethics Statement seems to suggest they went through the proper channels.

---

### Official Review · Reviewer_5nyV · 2025-11-07

**Soundness:** 2
**Presentation:** 1
**Contribution:** 3
**Rating:** 2
**Confidence:** 4

**Summary:**

The paper proposes IOHFuseLM, a two-stage framework for better IOH prediction. In the first stage, the author introduces Multi-Scale Trend Residual Diffusion Augmentation to augment the MAP sequence inputs via diffusion, and uses self-supervised training (recovering the masked segments of the MAP sequence) to pretrain this module. This self-supervised training is also highlighted as a multimodal learning process, as it incorporates cross-attention between the MAP sequence input and GPT-generated text descriptions. The second stage is a downstream task of predicting IOH based on the pretrained model from the first stage.  The proposed framework shows better performance on IOH prediction, compared with other benchmark models.

**Strengths:**

* The problem this work tries to address has high significance, and its solution is beneficial to its specific domain, i.e., providing insights into safe anesthesia.
 * The proposed framework achieves the best performance among other benchmark models studied in this work, and this state-of-the-art performance is validated on two datasets.
 * Visualization in Appendix F gives more insights into the model behaviors in the first pretraining stage.

**Weaknesses:**

The weaknesses are mainly in two aspects:

 * Methodological insufficiency/unclearness:
    * When preparing the text input, the additional GPT generation lacks justification. If the input text is just the template of "The age of patient is {age of patient}, gender is {gender of patient}, and the type of surgery is {surgery type of patient}", will the pretraining stage become easier or harder? A relevant issue is that although the appendix provides the prompt, there is no quality check on the outputs of GPT, for example, the evaluation from the domain experts on a small portion of the generated text to guarantee the overall correctness.
    * In the first stage, the author proposes a generation of sparse MAP series. This process reads like adding different levels of smoothness to the trend sequence, but lacks justification / empirical evidence on why this is necessary (ablation model 3 only ablates diffusion; line 240 gives high-level intuitions). Additionally, what makes the proposed sparse MAP trend sequence empirically or theoretically better than traditional 1D convolutional neural networks, or existing frequency-based time series methods such as the short-time Fourier transform?
    * Equation 9 confuses me. According to the visualization, augmentation extends known $x_{i, 1:l}$  to $x_{i, 1: l+t} $, where $x_{i, l: l+t} $ is predicted. But this equation shows the opposite, indicating that $x_{i, 1:l}$ is the generated sequence, while $x_{i, l: l+t} $ is known.
     * In the first stage, around line 298, the author states that "The resulting representations are concatenated with the series tokens and processed by a pretrained language model", and I fail to see any further explanation on why there is a pretrained language model and what it is pretrained for, besides a brief sentence in Appendix B saying "Our framework is based on GPT-2.". Additionally, the necessity of this pretraind LLM is not clear. Readers may guess that it utilizes the text information in the fused representation better than other approaches, but the claim needs to be clearly stated and justified in the main paper.
 * Inadequate empirical result:
     * There is no quantitative evaluation for the first stage besides Appendix F.2. A relevant issue is that when ablating the model, omitting any diffusion-based augmentation will lead to performance changes in both stages, but only the impact on the second stage is presented, while how the first stage masked segement recovery training is affected is unknown.

**Questions:**

Besides the weakness, this paper has some significant writing & notation issues:
 * Some symbols and terminologies are missing explanations. For example, MAP appears first in the introduction, and is explained in Section 3. Notations in equation 2 lack explanation, and readers can only guess that it might be age, gender, and surgery condition by checking the appendix.
 * There are critical typos. Equation 4 uses a term $x_{i.t+τ},$ which does not make any sense. Main plot figure 3 uses MTSDA  as the first stage model name, but it should be MTRDA.
 * Figure 3 is difficult to follow. For example, the color of the IOH Historical Series and IOH Augmented Series curves is almost the same, and it is not clear that the proposed model is a two-stage one, and the bottom left block is just a zoom-in of MTRDA in the first stage.

---

### Note · Authors · 2025-11-17

I have read and agree with the venue's withdrawal policy on behalf of myself and my co-authors.